# Maternal Immunization Using a Protein Subunit Vaccine Mediates Passive Immunity against *Zaire ebolavirus* in a Murine Model

**DOI:** 10.3390/v14122784

**Published:** 2022-12-14

**Authors:** Caitlin A. Williams, Teri Ann S. Wong, Aquena H. Ball, Michael M. Lieberman, Axel T. Lehrer

**Affiliations:** Department of Tropical Medicine, Medical Microbiology, and Pharmacology, John A. Burns School of Medicine, University of Hawaii at Manoa, Honolulu, HI 96813, USA

**Keywords:** filoviruses, breastmilk, protein subunit vaccines, mice

## Abstract

The Ebola virus has caused outbreaks in Central and West Africa, with high rates of morbidity and mortality. Clinical trials of recombinant virally vectored vaccines did not explicitly include pregnant or nursing women, resulting in a gap in knowledge of vaccine-elicited maternal antibody and its potential transfer. The role of maternal antibody in Ebola virus disease and vaccination remains understudied. Here, we demonstrate that a protein subunit vaccine can elicit robust humoral responses in pregnant mice, which are transferred to pups in breastmilk. These findings indicate that an intramuscular protein subunit vaccine may elicit Ebola-specific IgG capable of being transferred across the placenta as well as into the breastmilk. We have previously shown protective efficacy with these vaccines in non-human primates, offering a potential safe and practical alternative to recombinant virally vectored vaccines for pregnant and nursing women in Ebola endemic regions.

## 1. Introduction

Filoviridae is a family of negative-sense single-stranded RNA viruses capable of causing severe hemorrhagic fevers in human and non-human primates. Outbreaks caused by filoviruses can be large scale, as seen in the 2013–2016 epidemic of Ebola virus (EBOV) in West Africa and typically show high case fatality rates. EBOV and other filoviruses are highly transmissible and extremely virulent, penetrating many tissues and are recovered from several bodily excretions including, sweat, tears, and breast milk [1]. The Strategic Advisory Group of Experts tasked by the World Health Organization (WHO), evaluating the needs of pregnant and nursing women in the context of Ebola virus disease (EVD), has conditionally recommended the use of Ervebo^®^ as a preventative measure, citing the significant risks to the mother, fetus, and nursing infant through EVD. The extremely high case fatality rate and adverse pregnancy outcomes warrant the conditional use of a viral vectored vaccine [2]. An investigation into overall mortality in pregnant women with EVD which includes the 2013–2016 epidemic in West Africa shows a case fatality rate of 72% [3]. Case studies showed that nursing infants with infected mothers contracted EVD in 80% of the cases. Previous outbreaks have recorded extremely high case fatality rates for pregnant women and newborn infants, reaching 90% and 100%, respectively [4]. The WHO notes that the theoretical risk of harm to a fetus remains and calls for a vaccine method which reduces this potential risk. Our group has developed a recombinant protein subunit vaccine which induces protective immunity against live virus challenge in mice, guinea pigs, and non-human primates [5,6,7]. We previously demonstrated that EBOV glycoprotein (GP) purified using a Drosophila S2 expression system can elicit both active and passive protective immunity in mice when properly adjuvanted [6]. Here, we sought to leverage the existing mechanism of passive immunization in maternal and infant immunity to investigate the use of this protein subunit vaccine during gestation. Recombinant subunit vaccines are deemed safe for pregnant and nursing women as well as immunocompromised persons such as HIV-infected individuals [8]. The passive protection of neonates against infectious diseases has already been proven as a highly successful mechanism of immunity against respiratory and enteric pathogens [9,10], indicating that an effective EBOV vaccine may also facilitate passive protection in neonates prior to being able to build up immunity via active immunization.

## 2. Materials and Methods

### 2.1. Vaccination and Serum Collection

Female mice were immunized and paired with a non-immunized male on a schedule according to Table 1. One group of females was immunized but not paired with males to serve as non-pregnant controls. Immunizations occurred with a monovalent EBOV vaccine consisting of 10 µg of EBOV GP and 0.3 mg of CoVaccine HT^TM^. The immunogenicity, efficacy, and safety of this vaccine has been described in our previous studies [11]. Antigen and adjuvant were mixed in sterile PBS. Immunization was conducted by injecting mice intramuscularly into each hind leg with 50 µL of the aqueous vaccine preparation. Sera from pups were collected at week 3, 6, and 9 of life. Litters were divided into two groups each. One group was euthanized on the day of weaning (Day 21); blood was collected via terminal cardiac puncture to quantify antigen specific maternal antibody. The remaining juvenile mice were bled at week 6 and week 9 of life to assess the persistence of transferred maternal antibody. Blood was collected at week 6 via tail vein and via cardiac puncture at week 9. Whole blood was processed using a standard serum separation procedure and stored at −80 °C until use.

### 2.2. Mouse Cohorts

Female Swiss Webster mice were immunized and paired on two different schedules; the schedules were selected based on prior maternal antibody studies [12]. Females in group 1 (immediate) were immunized and paired with males on Day 0, while group 2 (delayed) was immunized on Day 0 and paired with males on Day 14. Both groups were immunized with a second dose on Day 21. Suckling mice were weaned after 3 weeks and monitored for up to 6 weeks post weaning.

### 2.3. Milk Collection and Delipidization

Milk was collected during the peak lactation period (lactation day 15–18) [13]. Milk was collected according to protocols laid out by the Prion Research Center at University of Colorado [14] with minor changes. Briefly, female mice were separated from pups to allow the engorgement of milk ducts for a period of no more than 3 h. Mice were anesthetized for the milk collection procedure using 5% isoflurane vapor. Anesthesia is administered using a vapor chamber and nose cone with the addition of gaseous O_2_ at a rate of 0.6 L/min. Once asleep, isoflurane was decreased to 2.5% and O_2_ was increased to 1 L/min, and the mouse was moved to the supine position. To prevent corneal damage from drying, eyes were lubricated with xylene ointment. To induce the secretion of milk, mice were given 2 IU/kg of oxytocin. Oxytocin was diluted into sterile PBS and administered at a dose of 2 IU/kg via intraperitoneal injection upon sedation. Milk was collected by manually palpating teats and using a p200 pipette for milk collection. Milk was refrigerated immediately after collection. Mice were monitored during the procedure and recovery period for signs of distress. Upon completion of the milking procedure, females were returned to their cage in the supine position and monitored until they rotated to the prone position. After rotation to the prone position, if no signs of distress were witnessed, mice were returned to their litter. Whole milk was delipidized by centrifugation as described previously [15]. Briefly, whole milk was centrifuged in a previously cooled centrifuge set to 4 °C at 16,000 rcf for 10 min. The aqueous layer was removed using a p10 pipette and stored at −80 °C until use.

### 2.4. Multiplex Assays

To determine total maternal IgG in pups, our in-house developed multiplex bead array assay was used. Antigen-specific immunoglobulin (maternal antibody) concentrations in mouse sera were measured as previously described with some minor alterations [11,16,17]. This assay used magnetic beads covalently linked to antigen. The antigens used in this assay are recombinantly expressed surface glycoproteins (GPs) of EBOV and *Sudan ebolavirus* (SUDV) as well as bovine serum albumin to serve as a negative control. Magpix beads were combined in 1X PBS + 1% BSA + 0.02% Tween 20 to make a master mix solution at a dilution of 1/200, corresponding to roughly 1250 beads per spectral region. Then, 50 µL of the microsphere suspension was added to each well of a 96-well microtiter plate. Serum was diluted at 1/1000 and 1/10,000. To quantify total IgG concentrations, a purified polyclonal mouse anti-EBOV GP IgG and an anti-SUDV GP IgG standard were serially diluted in a 1:2 dilution in the range of 8 ug to 1.95 ng/mL yielding a standard curve with a linear range of 125–1.95 ng/mL in the assay. Then, 50 µL of each dilution of serum was added to each well and incubated for 3 h on a plate shaker set at 650 rpm in a 37 °C incubator. Plates were washed three times with MIA buffer (1X PBS, 1% BSA, 0.02% Tween 20) using a magnetic plate separator (Millipore Corp., Billerica, MA, USA). Samples were subsequently probed with 50 μL of red-phycoerythrin (R-PE) conjugated F(ab’)2 fragment goat anti-mouse IgG specific to the Fc fragment (Jackson ImmunoResearch, Inc., West Grove, PA, USA) at 1/250 dilution and incubated for one hour at 37 °C with shaking at 650 rpm. Plates were then washed three times as described above, and 120 µL of Magpix drive fluid was added to each well to facilitate detection. The IgG subclass of maternal antibody and IgA content of pup sera were determined using goat anti-mouse polyclonal R-PE-conjugated antibodies (Southern Biotech, Birmingham, AL, USA) specific for IgG subclass IgG1, IgG2a, IgG2b, IgG2c, and IgG3 as well as IgA, at a 1/200 dilution. Plates were analyzed using the MAGPIX Instrument (MilliporeSigma, Watertown, MA, USA). Data acquisition detecting the median fluorescence intensity (MFI) was set to read a minimum of 50 beads per bead spectral region. Antigen-coupled beads were recognized and quantified based on bead spectral region and signal intensity, respectively. The total IgG concentrations were calculated by interpolating the standard curve using Prism 8 (Graphpad Software, San Diego, CA, USA) and multiplying by the dilution factor. The limit of quantification was set as the lower limit of detection by the standard, 1.95 ng/mL. For assays reporting MFI readings directly, the assay cutoff was determined by calculating the mean fluorescence of the assay negative control plus three standard deviations, using Microsoft Excel (2011). Data were graphed using Prism 8 (Graphpad Software) and Microsoft Excel 2011. These data were compared to control animal data we previously obtained (“historical controls”) from our published results using the recombinant EBOV protein subunit vaccine in mice [6].

### 2.5. Surrogate Neutralization Assays

Based on the high correlation between WT-EBOV plaque reduction neutralization assays and the rVSV-EBOV-GFP fluorescence reduction neutralization test (FRNT) leading to the subsequent recommendation for use in clinical trials [18], a microneutralization assay using rVSV-EBOV-GFP was designed based on previously established protocols [19,20,21]. Surrogate neutralization assays were carried out using recombinant vesicular stomatitis virus expressing ebolavirus glycoprotein as the virion surface protein (VSV-EBOV-GFP). The VSV-EBOV-GFP encodes the EBOV-Kikwit GP instead of VSV-G, and the GFP (green fluorescent protein) ORF was inserted between the EBOV GP and VSV-L genes [22]. rVSV-EBOV-GFP was recovered from this plasmid as described previously [23] and was generously gifted by Andrea Marzi (Laboratory of Virology, NIAID, Hamilton, MT, USA). Vero cells were plated in Dulbecco’s Modified Eagle Medium (DMEM) supplemented with 2% fetal bovine serum and plated on a clear flat-bottom 96-well plate at a concentration of 25,000 cells per well. Serum from mouse pups was heat inactivated at 56 °C for 45 min and diluted to 1:10 followed by 8 serial 1:2 dilutions. Serum dilutions were then mixed with 3 × 10^4^ PFU/mL of recombinant vesicular stomatitis virus expressing EBOV GP and GFP (rVSV-EBOV-GFP) and incubated for 1 h at 37 °C (5% CO_2_). Serum/virus mixtures were added to Vero cells, 50 µL per well, and incubated for 16 h at 37 °C (5% CO_2_). Sixteen hours after infection, supernatant was removed from each well, and cells were fixed by adding 50 µL of 1% paraformaldehyde in PBS. Plates were incubated for 15 min at room temperature and then washed three times with sterile 0.1X PBS. Fixed cells were stored in 50 µL of PBS at 4 °C until reading. Plates were read using the CTL-ImmunoSpot^®^ S6 Universal Core Analyzer. GFP positive cells were visualized and counted using the scan and basic count: inverted normal functions of ImmunoSpot Fluoro-X Suite Software. The spot counts in wells with serum dilutions were normalized to the counts in virus-only control wells with the following equation: (virus only count–experimental count)/(virus only count) × 100%. Data were analyzed using Prism 8 and Microsoft Excel 2011. Neutralization titers (NT_50_) titers were determined as a cutoff of the reciprocal of the highest dilution able to neutralize 50% or more of virus as compared to the virus-only control wells. All samples and controls were run in triplicate.

### 2.6. Statistical Analysis

Statistical analysis of the variation between antigen-specific IgG concentrations between the milk and sera of individual mice was conducted using Student’s *t* tests in Prism 8 (Graphpad Software). Statistical analysis to determine the difference in variation between mothers throughout the immunization schedule and variation between litters was conducted using a two-way ANOVA in Prism 8.

## 3. Results

### 3.1. Antigen-Specific IgG in Mothers and Pups

Non-pregnant female Swiss Webster mice were immunized with two doses of 10 µg of EBOV GP adjuvanted with 0.3 mg of CoVaccine HT^TM^. Mice were immunized at day 0 and day 21 of the study. To investigate potential differences in the transfer of maternal antibody, females were either immediately paired with a male on day 0 after primary vaccination or paired two weeks later on day 14 (Figure 1a). Blood samples were collected according to the immunization schedule; milk was collected during peak lactation, day 15–18 of lactation (Figure 1a). Immunization with 2 doses of vaccine consistently elicited high concentrations of antigen-specific IgG in serum of mouse mothers (Figure 1b). There was no difference in the antigen-specific IgG response between pregnant mice and non-pregnant controls. Female mice in this cohort responded on par with historical controls, female Swiss Webster mice immunized with 10 µg of EBOV GP and lyophilized CoVaccine HT^TM^ (Figure 1b). Non-vaccinated (naive) negative historical control mice do not demonstrate a detectable EBOV immune response (data not shown [6]). Blood was collected from pups at week 3, 6, and 9 of life. Antigen-specific IgG concentrations were similar between dams in both groups after each vaccine dose, and they were detectable in milk samples as well as pups for the entire study (Figure 1c). In the immediate pair group, antigen-specific IgG was concentrated in the pups and found in higher concentration at week 3 after birth (weaning) compared to the mother’s sera (*p* = 0.001) and milk at peak lactation (*p* = 0.02) (Figure 1c). In the delayed pairing group, the concentration effect of highly efficient transfer of maternal IgG is not detected in the week 3 pups, and concentrations are not higher in the pups as compared to the mother’s sera and milk from the peak lactation. This observation was expected as the maternal concentrations differed due to time since immunization (Figure 1c). In both groups, antibody concentration waned in the weeks post weaning. In the immediate pairing group, the degree of waning between week 3 of life and week 9 of life was statistically significant (*p* = 0.0008). Considering the antigenic similarities between *Sudan ebolavirus* GP (SUDV GP) and EBOV GP, some level of cross-reactivity is expected. We measured cross-reactive IgG in pups at peak lactation in the dams and at weeks 3, 6, and 9 of life in the pups to determine if cross-reactive IgG is transferred to offspring at similar levels. Cross-reactive antibody concentrations were similar to those of the EBOV GP-specific concentrations, with minor waning seen through weeks 6 and 9 of life in the pups in the immediate pairing group (Figure 2).

### 3.2. Immunoglobulin Isotype and IgG Subclass

Antibody isotype and subclass were also measured by a magnetic bead multiplexed immunoassay (MIA). Serum IgA was not detected above the background in either vaccination/pairing group, indicating a lack of antigen-specific IgA in immunized dams, their milk, and their offspring (Figure 3). A diverse population of IgG subclasses is detected in the serum and milk from mothers and serum from pups. IgG1 is transferred at the highest concentration followed by IgG2a and IgG2b. IgG1, IgG2a, and IgG2b are detected in the highest concentration in the week 3 pup sera as well as in the milk. To a lesser extent, IgG2c and IgG3 are transferred to pups. The amounts of IgG1 and IgG2a seem to be different between the mother and the pups. In the immediate vaccinee and pair group, pups have a higher concentration of IgG1 and IgG2a than mothers. This is corroborated by the overall higher IgG detected in pups at this timepoint. IgG2c and IgG3 are found in relatively similar concentrations across mother’s serum, milk, and week 3 pup sera. As expected, IgG concentrations decreased; thus, the week 9 pup sera showed lower antigen specific IgG subclass levels compared to the week 3 pup sera (Figure 3a). In the delayed pairing group, IgG1 and IgG2a are found in similar concentrations between mothers and week 3 pups (Figure 3b). IgG2b, IgG2c, and IgG3 are found in the highest concentration in the mothers, which is followed by pups at week 3 of life (Figure 3b).

### 3.3. Neutralization of rVSV-EBOV-GFP

A virus neutralization assay utilizing recombinant vesicular stomatitis virus encoding EBOV GP as well as GFP was conducted to determine the potency of recombinant viral neutralization by maternal IgG transferred to pups. Neutralizing IgG has been associated with the protective efficacy of monoclonal antibody therapy used to treat EVD [24]. In this assay, serum samples from pups in each litter were pooled to analyze neutralization capacity (neutralization titer) at week 3 and at week 9 of life, which is defined as the highest serum dilution that reduced GFP positive cells by at least 50%. Serum at week 3 of life had some capacity for neutralization (Table 2, Figure 4) in all groups of the immediate vaccinate and pair cohort; however, in the delayed pairing cohort, serum pools from most litters were unable to neutralize virus (Table 2, Figure 4). The virus-neutralizing serum dilution factor was between 40 and 160 for all animals in the immediate vaccinate and pair group (Table 2). While the neutralizing titer decreased by two-fold between week 3 and 9 for litters CW4 and CW6 (Table 2), no changes were seen in titers for litters CW1 and CW2 (Table 1). Inhibitory titers in the delayed pairing group were below the starting dilution for litter CW7 and CW9 (Table 2); however, litters CW10, CW11, and CW12 were able to neutralize virus at week 9 of life (Table 2). Litter CW10 and CW11 had 2X the neutralization potency compared to other litters in the same cohort; at week 3, the 50% inhibitory titer was 1:80. Statistical analysis with Tukey’s Multiple Comparisons test was used to compare the virus neutralizing titers between each group at each timepoint. Group 1, the “immediate vaccinate and pair group”, was significantly different than the delayed group at each timepoint (Figure 4b).

## 4. Discussion

As new vaccines become available in EBOV endemic regions, it is pertinent to understand the potential of preventing infant mortality through maternal immunization. Case fatality rates of children are extremely high, reaching between 80 and 100% for neonates [3]. A vaccine which can induce protection in both mother and child would decrease the disease severity in this population in EBOV-endemic regions. We have previously demonstrated the protective efficacy of our protein subunit vaccine in mice against mouse-adapted EBOV challenge as well as in non-human primates using wild-type EBOV isolated from human infection [5,6]. Furthermore, we demonstrated that protective efficacy can be reached through the passive immunization of mice, using donor serum and T cells elicited by the same formulation used in the work presented here [6]. In the present study, we demonstrate that this protein subunit vaccine can induce robust antigen-specific antibody in pregnant mice, which is further transferred to their pups. Moreover, pregnancy in adult female mice did not affect the development of EBOV GP-specific IgG responses. Indeed, the MFI of EBOV GP-specific IgG is comparable to that of non-pregnant adult female mice as indicated in Figure 1; these findings are comparable to our previously published work in immunized mice using this vaccine formulation [6]. Most importantly, this work demonstrates that systemically generated IgG, produced from intramuscular exposure to EBOV GP antigen, results in a high concentration of GP-specific IgG in the milk which is then transferred to suckling mouse pups. Our data show that maternal IgG persists in the periphery of pups for at least 6 weeks post weaning. Whether or not this would interfere with immunizations given to these pups is unknown; however, the literature suggests it may not [25]. We demonstrated that the window for the optimal transfer of vaccine-induced maternal IgG is when peak titers coincide with peak lactation. Regarding human immunizations, this indicates that vaccination in the third trimester, coinciding with peak placental transfer, and subsequent IgG transfer via colostrum would be ideal for inducing passive immunity in neonates.

A key finding was the lack of development of antigen-specific IgA. The MIA did not detect anti EBOV GP IgA above background levels, indicating a lack of class switching. This could be due in part to the route of immunization or the vaccine characteristics. An intramuscular inoculation induces the development of immunoglobulin distal to any mucosal tissue. The development of antigen-specific IgA might require a mucosal route of inoculation in the dams or different adjuvantation. However, IgA in milk does not cross the gut epithelial lining and therefore does not enter into the periphery and may not play an active role in preventing EBOV acquisition. At this time, the route of infection in infants is not well defined due to the frequent exposure to various bodily fluids from the mother, including sweat, breastmilk, and saliva [3].

IgG transferred to suckling pups consisted of multiple subclasses. IgG1 was present at the highest concentration and IgG3 was the lowest. This is consistent with the relative levels of these subclasses in the mother’s serum, indicating no preference in subclass transfer by mouse FcRn. However, this differs from humans where FcRn exhibits preferential transfer of IgG1 followed by IgG3, IgG4, and IgG2 [26]. Other studies have previously reported that in mice, maternal IgG transfer is facilitated via FcRn rather than FcyR [27,28], which is consistent with human transfer of maternal antibody. Other investigators report preferential transfer of IgG3 [27]; in contrast, the serum from suckling pups in our study showed more IgG1 than IgG3. This could be due to the relatively lower concentration of IgG3 relative to IgG1 in the mothers. The approximate ratio between IgG1 and total IgG content was calculated by taking the ratio of MFI for IgG1 detected by an anti-mouse IgG1 antibody and MFI detected by an anti-mouse IgG antibody, which in our study yielded a ratio of 0.7 (data not shown). In mice, the IgG1, IgG2a, IgG2b, and IgG2c subclasses are associated with functions such as complement activation and opsonization, indicating a potential for polyfunctionality in the periphery of neonates. The subclass transfer of maternal antibody may be key, as the different IgG subclasses carry out different functions. In mice, the IgG1 subclass is associated with a Th2 response, whereas the IgG2a/c is associated with FcyR-mediated responses, and its production is upregulated by IFNγ. IgG2b and IgG3 are associated with a T-cell independent response and activation of the complement cascade as well as FcyR binding, although this binding is more moderate compared to the IgG2a binding [29]. Interestingly, the maternal antibody transferred to pups did not appear to neutralize virus consistently between litters of the same vaccination regimen. In the “immediate pairing” group, two litters had neutralizing titers of 1:80, while the other litters had neutralizing titers of 1:40. Most of the pups in the delayed pairing group were outright unable to neutralize virus, which was most likely due to different kinetics in antibody titers associated with the delayed immunization. This is consistent with the overall antigen-specific IgG concentrations in which the immediate vaccinee and pair group had the highest amount of maternal antibody at week 3 compared to other timepoints as well as the delayed pairing cohort. A high titer mother and a low titer mother also demonstrated variability in neutralization potency, indicating that the capacity of neutralization is likely dependent on the ability of the mother to rapidly generate virus-neutralizing titers. In this study, we utilized a vaccine formulation which we previously demonstrated elicited protective efficacy against mouse adapted EBOV; here, we substituted viral challenge in a BSL4 facility for surrogate neutralization assays. Konduru et al. have previously described this assay platform, which is strongly correlated with neutralization assays conducted with wild-type EBOV in BSL4 facilities [18].

These studies demonstrate that a protein subunit vaccine can elicit a robust humoral response in pregnant Swiss Webster mice, which is efficiently transferred to pups. We demonstrated that suckling pups are capable of absorbing and concentrating maternal IgG in their periphery, resulting in comparable antigen-specific IgG concentrations in immunized mothers and unvaccinated pups. Our findings indicate that the maternal antibody transferred to pups is consistent with the isotype and subclass and overall Ig concentration of the mother. Our vaccine elicits a diverse IgG response; however, immunogenicity results from this model did not demonstrate an IgA response. Overall, our results document an antigen-specific and functional humoral response in suckling pups, which lasts at least several weeks post weaning. IgG remains present in both transitional and mature human milk, indicating it has continuous potential for functionality within a nursing infant; how efficacious this transfer is remains largely unknown as the FcRn itself remains on the surface of epithelial tissues into adulthood. These mechanistic and qualitative approaches could aid in developing immunotherapies which target mother–child nursing dyads in the context of mitigating filovirus disease and could further aid in developing strategies for leveraging maternal antibody as an IgG therapeutic.

## Figures and Tables

**Figure 1 viruses-14-02784-f001:**
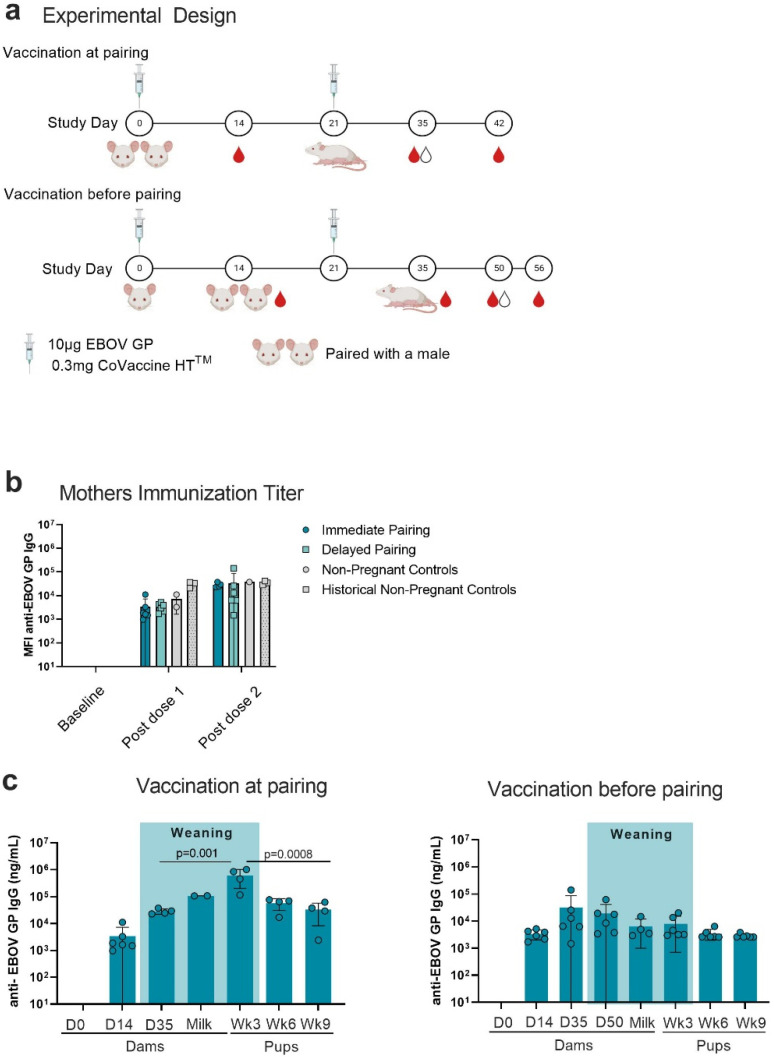
Immunization schedule and antigen-specific IgG concentrations. (**a**) Immunization schedule. Swiss Webster females were immunized with 10 µg of purified EBOV GP adjuvanted with 0.3 mg of CoVaccine HT^TM^ on day 0 and day 21. Female mice were vaccinated on day 0 and then either paired with non-vaccinated males on day 0 (“vaccination at pairing”) or paired with non-vaccinated males on day 14 (“vaccination before pairing”). Blood was collected from immunized dams on day 0, 14, 35, 42 in the immediate pair group, and on day 0, 14, 35, 50, 56 in the delayed pair group. (**b**) Antigen-specific IgG detected in the mothers. Anti-EBOV GP antibody was measured in mother’s sera at day 0, 14, and 35. Historical controls are female Swiss Webster mice immunized with a lyophilized version of the aqueous vaccine used in this study. (**c**) Anti- EBOV GP antibody was measured in mother’s serum or milk during peak lactation (between day 15 and 18 post-partum) and in pup sera at weeks 3, 6, and 9 of life. Each data point represents an individual mouse litter. Statistical analysis was conducted using a 2-way ANOVA with multiple comparisons in GraphPad Prism. Figure Created with BioRender.com (20 November 2022).

**Figure 2 viruses-14-02784-f002:**
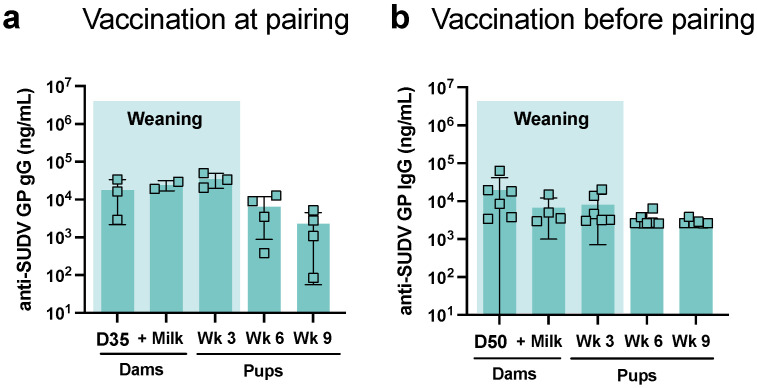
Cross-reactive antibody concentrations. Female mice were vaccinated on day 0 and then either paired with non-vaccinated males on day 0 (“vaccination at pairing”) or paired with non-vaccinated males on day 14 (“vaccination before pairing”). Blood was collected from immunized dams for serology. SUDV GP specific IgG was measured in both groups. (**a**) Anti-SUDV GP antibody was measured in mother’s serum at day 35 or (**b**) on day 50. Antibody in milk was measured during peak lactation between day 15 and 18 post-partum. Anti-SUDV GP antibody was measured in pups at weeks 3, 6, 9 of life. Each data point represents an individual mouse litter.

**Figure 3 viruses-14-02784-f003:**
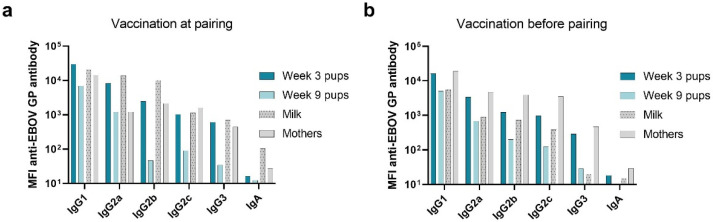
Antigen-specific isotype and IgG subclass analysis. Immunoglobulin isotype and subclass was determined in immunized mothers and in milk at peak lactation and in pups at week 3 and 9 of life. Median fluorescence intensity of antigen-specific isotype and subclass are reported for (**a**) immediate pairing litters and (**b**) delayed pairing litters.

**Figure 4 viruses-14-02784-f004:**
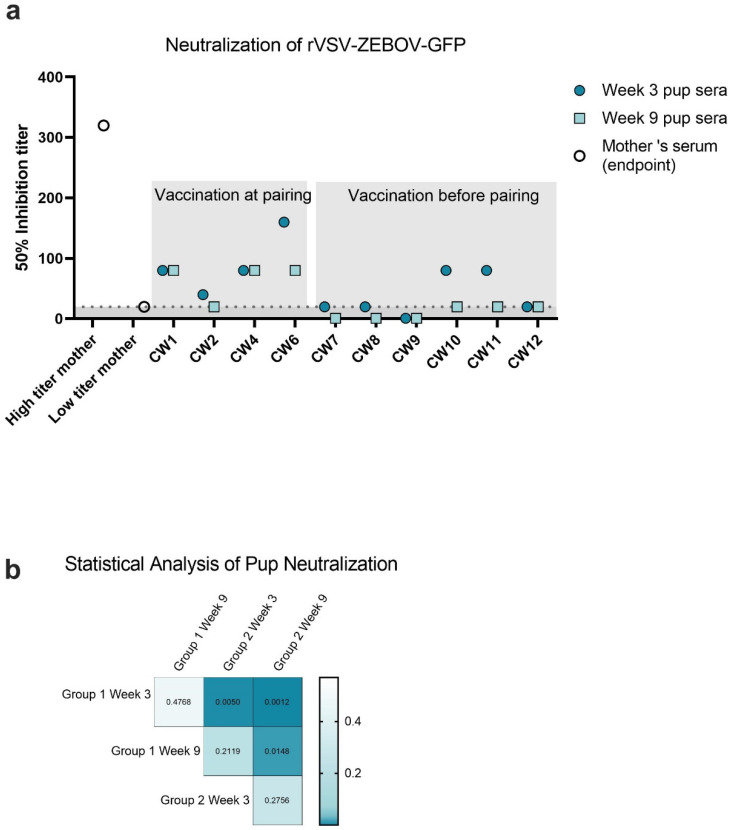
Neutralization of rVSV-EBOV-GFP. Neutralization capacity of maternal antibody was measured by counting GFP positive Vero cells after 16 h culture with diluted serum and virus. Wells were seeded with 2.5 × 10^4^ cells and 50 µL of serum-virus mixtures containing 750 PFU of rVSV-EBOV-GFP were added. (**a**) The 50% inhibition titers of two selected mothers (high titer and low titer) at week 9 and pups at weeks 3 and 9 of life. Serum samples were collected via cardiac puncture. (**b**) Heatmap of adjusted *p* values comparing neutralization titers for the group vaccinated at pairing (group 1) and the group vaccinated before pairing (group 2), calculated using Tukey’s Multiple Comparisons Test.

**Table 1 viruses-14-02784-t001:** Mouse cohorts.

Pairing	No. of Female SW	No. of Litters	No. of Pups at Week 3	No. of Pups at Weeks 6 & 9	Total
Immediate	6	4	18	23	41
Delayed	6	6	30	37	67

**Table 2 viruses-14-02784-t002:** Pup Serum Neutralization of rVSV-EBOV-GFP.

Group	Litter	Week	50% Inhibitory Titer	n
Immediate Pair	CW1	3	80	4
		9	80	4
	CW2	3	40	6
		9	20	7
	CW4	3	80	2
		9	80	8
	CW6	3	160	6
		9	80	4
Delayed pair	CW7	3	20	7
		9	<20	8
	CW8	3	20	6
		9	<20	6
	CW9	3	<20	6
		9	<20	8
	CW10	3	80	4
		9	20	5
	CW11	3	80	3
		9	20	5
	CW12	3	20	4
		9	20	5

## Data Availability

The raw data supporting the conclusions of this article will be made available by the authors, without undue reservation.

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
