# Peer review of "Maternal Immunization Using a Protein Subunit Vaccine Mediates Passive Immunity against Zaire ebolavirus in a Murine Model"

_viruses, 2022, doi:10.3390/v14122784_

Round 1

Reviewer 1 Report

The article By Williams et.al has shown that subunit vaccine against Ebola virus can elicit humoral response in pregnant mice and can be passed to pups via breast milk, here they have compared the immediate and delayed effect of vaccination and found that they had higher antigen specific titers higher in immediate pairing group at week 3 than the other one. However in the article they have used only one formulation of vaccine and concentration throughout the study. In the article they haven’t shown the survival rate or any effect to pups or vaccinated female due to vaccination. In the results it’s shown that they have used non pregnant females as control, but no explained in materials and methods. No unvaccinated controls were used in the rest of studies.

In the introduction its written EVD, but not given explained as Ebola virus disease anywhere in the content

In page 2 line 3rd paragraph its written lactation day as 15- 18 where as in page 4 Results its 15-17, please uniform

In page 4 Material and methods its written 10 uM PBS please recheck and confirm (10 uM or ul)

In figures- please unify the titles in all figures (size and font)

Fig 2 no complete description in the legend

Page 7 – line 1 and 2 no complete sentence

Fig 4b- no tittle

Page 8- In figure legend line 3 Infected with 750 PFU/ml of recombinant vesicular stomatitis virus expressing EBOV GP and GFP (rVSV-EBOV-GFP)

Please add further reference and summary/conclusion to study

Reviewer 2 Report

The manuscript by Williams et al. presents data on investigation of maternal immunization with a subunit vaccine against Ebola virus in a murine model. The authors found that maternal immunization of pregnant mice results in induction of antibodies specific for Ebola virus GP protein in both vaccinated dams and pups. The authors found that the antibodies cross-reacted with Sudan virus GP protein and were capable to neutralize recombinant VSV-Ebola GP chimeric virus. In addition, the authors determined isotypes and subclasses of Ebola-specific IgG. 

The manuscript is well written, and the results are clearly presented and well discussed. 

Major

1. The work does not include challenge experiment, which can be performed with mouse-adapted Ebola virus. Mouse dams can be vaccinated with almost any protein, which will induce some antibody response detectable in dams and pups. Even if mouse challenge is included, the results cannot easily predict protection in human due to difference in the disease caused by Ebola virus in mouse and human. Without determining protection against challenge with Ebola virus, the data included in the manuscript is inconclusive. There are multiple BSL-4 labs in the United States, which can run the challenge study.

2. The Ebola virus-neutralizing antibody response is determined using the surrogate system, which in many cases does not predict neutralization of authentic Ebola virus. While the surrogate system used by the authors is good for screening purposes, it should not be used for studies when the protection cannot necessarily be expected. 

Minor

1. Section 3.1. “females were either immediately paired with males on day 0…” – does that mean that on day 0 mice were given the first vaccine dose and then paired on the same day? Same question for the group which was paired after the second vaccine dose. This needs to be clarified in the text and in figure 1a.

2. Some labels (legends) in the body of figures 1a, b, c are too small and unreadable. 

3. Introduction, 2013 – 2016 Ebola outbreak. Better to describe this as “epidemic”.

Round 2

Reviewer 2 Report

My major criticism was that only antibodies in pups, but not the protection, was analyzed. This criticism has not been addressed.

Without the protection data, this manuscript should not be accepted by Viruses.

Author Response

In agreement with the reviewer's concern and the editor's concurrence, we revised the title of the manuscript to reflect the fact that we did not test protection, but report transfer of passive immunity to mouse pups. Throughout the manuscript we also deleted or revised statements to reflect this fact to not mislead readers.